# Stop Guessing When to Stop Testing: Efficient Model Evaluation with Just Enough Data

## 1 Introduction

The rapid advancement of large vision-language models (VLMs) and language models (LLMs) has spurred the creation of numerous benchmarks to assess their capabilities (Li & Lu, 2024; Chang et al., 2023). However, processing high-resolution images, handling large contexts, comparing performance across multiple datasets, and utilizing expensive metrics like LLM-as-Judge have drastically increased evaluation costs (Zhao et al., 2024; Perlitz et al., 2024). Current evaluation practices, typically employing fixed-size benchmarks, are inherently wasteful, continuing to the predetermined sample size even when the outcome is statistically clear. While practitioners often reduce costs by using fewer samples (Perlitz et al., 2024; Polo et al., 2024; Fogliato et al., 2024; Zhao et al., 2024), these heuristic approaches lack statistical guarantees.

Critically, fixed-size approaches, whether high-cost and precise or low-cost and imprecise, fail to align the evaluation effort with the evaluation's objective. Debugging a model may only require a inexpensive coarse approximation, while definitively determining a superior model among close contenders demands sufficient data to achieve statistical significance.

In this work, we propose a statistically grounded solution: an *adaptive evaluation* framework based on sequential testing. Rather than enforcing a fixed sample size, adaptive evaluation stops when the practical and statistical needs are met. Thus, users explicitly define their needs, and the method ensures that evaluations are neither underpowered nor excessive, effectively balancing reliability and efficiency. In addition, when efficiency is prioritized, users are fully aware of what was lost in terms of statistical power, allowing them to make informed decisions rather than rely on guesswork. This makes adaptive evaluation not only *efficient* but also *transparent*.

Our contributions are: 1) A call to eschew fixed-size evaluation. 2) A framework to optimize sample efficiency while maintaining statistical reliability through the adoption of sequential testing.[1] 3) An analysis of efficiency-reliability trade-offs in single-model scoring, pairwise comparisons, and ranking tasks.

## 2 Use Case Examples

To demonstrate the versatility and impact of adaptive evaluation, we highlight three diverse use cases where our approach provides clear advantages over traditional fixed-sample methods. We show the empirical results of our method in § 7 and show usecase experiments in §7.3.

**Compute-Constrained Evaluation with Statistical Guarantees.** Many practitioners must evaluate models under limited compute and time budgets. Traditionally, this means arbitrarily subsampling benchmarks (or worse, skipping some datasets altogether (Perlitz et al., 2024)), sacrificing reliability in an unknown way. With adaptive evaluation, users can stop evaluation early, without compromising reliability.

**Meaningful Change Achieved** In production, teams must repeatedly decide whether to replace an existing model with a new one. However, not all improvements are meaningful; gains of 0.1 points – even if statistically significant – may not justify deployment. Adaptive evaluation ensures that comparisons stop as soon as the observed difference is statistically and practically significant, preventing both over-evaluation and premature conclusions.

---

[1] We plan to release the code upon acceptance of this paper.

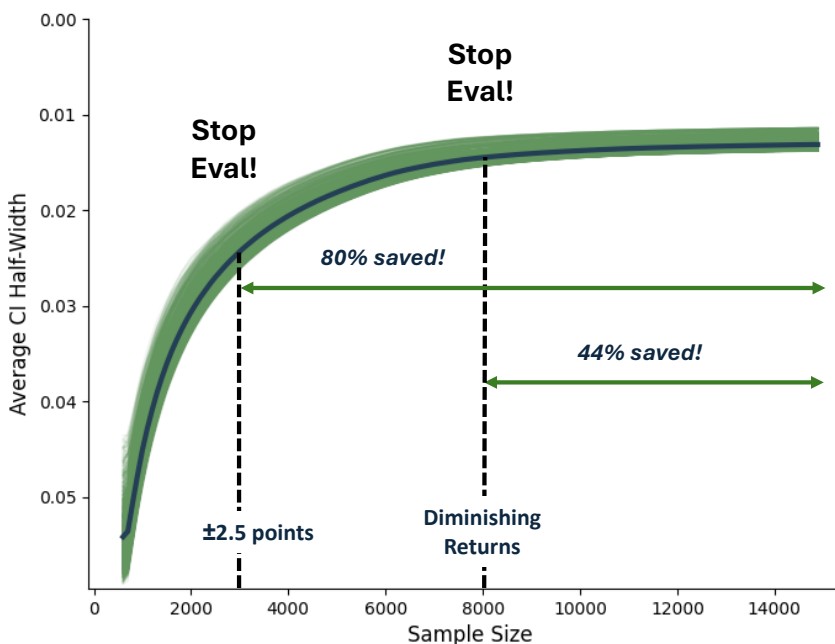

Figure 1: Half-width of the confidence interval as a function of sample size for 206 models in the Open VLM Leaderboard benchmark, averaged over 10 random seeds. As sample size increases, the CI narrows, reducing uncertainty in model performance estimates. Two stopping strategies are illustrated: (1) stopping when the CI reaches $\pm 2.5$, saving 80% of the evaluation cost, and (2) stopping when diminishing returns plateau, reducing cost by 44% while sacrificing only 0.132 points in precision. Our framework can detect and stop evaluation based on such rules, ensuring both statistical rigor and evaluation efficiency.

**Model Development: Efficient Candidate Model Selection.** Modern model pre and post training pipelines often produce hundreds or thousands of candidate models, many of which are redundant or under-performing. As such projects are often under tight time constraints, evaluating all models fully is impractical, and thus the need for rapid assessment makes efficiency crucial. Adaptive evaluation enables early detection of both promising and faulty checkpoints, terminating evaluations only when necessary, dramatically reducing computational waste and saving time (see §7.3).

The above examples demonstrate the versatility of adaptive evaluation, aiding single model decisions and full ranking, saving compute, time, annotation effort, mistakes due to unreliability and more.

## 3 RELATED WORK

**Efficient Evaluation.** Previous works have explored the use of small subsets of benchmarks for efficient evaluation (Choshen et al., 2024), analyzing their impact on reliability (Perlitz et al., 2024; Maynez et al., 2023). Other approaches have proposed intelligent sampling methods to maintain reliable results (Polo et al., 2024; Zhang et al., 2024; Vivek et al., 2024). However, many of these methods rely on past statistics, requiring full benchmark runs on multiple models and assuming future models will follow similar distributions, which limits their applicability in dynamic evaluation settings (Zhang et al., 2024; Vivek et al., 2024). Some other methods predict model scores (Zhao et al., 2024), or improve reliability without providing guarantees (Fogliato et al., 2024). In contrast, our method offers strict statistical guarantees without needing to run models on the full benchmark to gather past statistics. The closest work to ours uses sequential testing to stabilize performance estimates in reinforcement learning (RL) algorithms, where multiple runs with different seeds are required for statistical significance (Mathieu et al., 2023). Unlike their RL-specific focus, our framework prioritizes sample efficiency and generalizes to diverse tasks, enabling adaptive evaluation with user-defined stopping criteria for model comparison, ranking, and benchmarking.

## 4 DEFINITIONS AND BACKGROUND

We propose a method to stop given certain criteria and account for peeking into the data in the reported statistics. In this paper, we focus on the general case of multi-dataset benchmarks, which are increasingly common in general-purpose model assessment and particularly sensitive to this trade-off (as shown in § 7). Formally, let a benchmark consist of $n$ datasets:

$$D = \{D_1, D_2, ..., D_n\},$$

where each dataset $D_i$ contains $m_i$ examples $e_{ij} \in \mathcal{E}_i$

$$D_i = \{e_{i1}, e_{i2}, \ldots, e_{im_i}\},$$

and has an associated scoring function $S_i : \mathcal{E}_i \to [0, 1]$, where higher values indicate better performance. We define $S_i^A$ as the application of $S_i$ on the outputs of model $A$. The overall benchmark score for model $A$ is given by

$$S_A(D) = \frac{1}{n} \sum_{i=1}^{n} \left( \frac{1}{m_i} \sum_{j=1}^{m_i} S_i^A(e_{ij}) \right). \tag{1}$$

This mean-of-means formulation ensures equal weighting across datasets, regardless of individual sizes.

A typical evaluation involves finding the score of a model $A$ or comparing $A$ against a baseline $B$ on benchmark $D$, where the observed performance difference is

$$\hat{\delta}_{A,B}(D) = S_A(D) - S_B(D). \tag{2}$$

If $\hat{\delta}_{A,B}(D) > 0$, model $A$ appears better than $B$, with a larger $\delta$ indicating a greater performance difference. However, this observed difference could be due to chance. Put simply, the reliability of an evaluation is how much we can trust that the observed difference reflects true model performance rather than measurement noise.

Broadly speaking, the more examples we use, the more confident we can be in our conclusions. However, the number of available examples in a benchmark might be excessive, or not enough. This creates a fundamental challenge: how to control the trade-off between reliability and efficiency in model evaluation.

### 4.1 QUANTIFYING RELIABILITY: STATISTICAL SIGNIFICANCE

To enable adaptive evaluation, we first need a formal definition of reliability. The standard statistical framework for this comes from the theory of hypothesis testing and confidence intervals Dror et al. (2018).

Given a benchmark $D$, we assess the probability that an observed difference in performance between models $A$ and $B$, $\hat{\delta}_{A,B}$ (2) could occur by chance under the null hypothesis ($H_0$). This is formulated as the following hypothesis test:

$$H_0 : \delta_{A,B}(D) = 0,$$
$$H_1 : \delta_{A,B}(D) > 0.$$

Here, $\delta_{A,B}(D)$ represents the true performance difference. A p-value is given by the probability to observe a difference at least as extreme as the one observed, under the assumption that $H_0$ is true. If the p-value is greater than a predefined threshold $\alpha$, we reject $H_0$ and conclude that model $A$ is better than $B$ at statistical significance level $\alpha$.

The results of hypothesis testing are influenced by sample size, effect size, and the significance level ($\alpha$). Larger sample sizes help reduce noise, making it easier to detect true differences. A larger effect size, or true difference between models, also makes the difference more detectable. The significance level determines the threshold for rejecting $H_0$, which directly impacts the test's power and the reliability of conclusions.

Confidence intervals offer a complementary view by quantifying uncertainty, providing a likely range of plausible values for the true performance difference between models. For example, given

$\alpha = 0.05$ a CI $[L, U]$ suggests that the true performance difference or the model's scores lie within this range with 95% probability. However, repeatedly checking (and acting upon) results during data collection, a practice known as "peeking," invalidates the standard interpretation of p-values and confidence intervals.

# 5 PROPOSED ADAPTIVE FRAMEWORK

Several key capabilities are necessary to enable efficient and statistically sound adaptive evaluation.

The first one (see §5.1) is providing near real-time insights into the model's performance. Naively checking the performance at multiple stages of the evaluation can lead to inflated error rates and reduce reliability, since the probability of obtaining a false positive increases.

The second one (see §5.3) is to adapt the evaluation process as more data is collected. This includes implementing stopping rules that help determine when to halt the evaluation based on the accumulated evidence. By using these rules, we can stop testing once we gathered that necessary information rather than when we run out of resources.

In this section, we introduce a sequential testing-based framework for adaptive testing with standard criteria for adaptive stopping, and discuss how different stopping rules align with evaluation objectives.

## 5.1 MAINTAINING VALIDITY WITH SEQUENTIAL TESTING

We formalize adaptive evaluation using the group sequential testing framework with the Pocock spending function Pocock (1977). Rooted in classical statistical analysis (Wald, 1945b; Jennison & Turnbull, 1999a; Lan & DeMets, 1983; O'Brien & Fleming, 1979), this approach has been widely applied in fields like clinical trials and quality control Jennison & Turnbull (1999b). Sequential testing addresses the reliability-efficiency tradeoff, enabling data-driven decisions on when to stop or continue testing based on evolving needs.

While other sequential analysis methods and spending functions, such as *SPRT* method Wald (1945a) and *O'Brien-Fleming* spending function O'Brien & Fleming (1979), are available, we choose the Pocock spending function for demonstration purposes, as it provides a well-balanced framework particularly suited for batch-based model evaluation. Though this and any sequential analysis method introduces a modest power reduction and slight computational overhead, its efficiency gains far outweigh these costs. Next, we present the mathematical formulation of group sequential testing adapted for model evaluation.

## 5.2 GROUP SEQUENTIAL TESTING

Formally, group sequential methods partition the evaluation process into $t$ stages. At each stage $k$, data is collected in a batch of size $b$, resulting in a cumulative sample size $N_t$ defined as

$$N_k = kb, \quad k = 1, \ldots, t.$$

The test statistic $Z_k$ is computed using all data accumulated up to stage $k$.

The choice of $t$ significantly influences the testing process. A larger $t$ enables more frequent interim analyses, potentially allowing earlier stopping decisions, but it requires a more samples to account for multiple testing (i.e., repeated "looks" at the data). In practice, we find it not to be an issue as long as the batch is reasonably sized (e.g., not 1).

For pairwise comparison of models $A$ and $B$ using an evaluation metric $S$ (see Eq. 1), the test statistic at stage $k$ is defined as

$$Z_k^{AB} = \frac{S_k^A(D) - S_k^B(D)}{\sqrt{\hat{\sigma}_k^2 \cdot (1/N_k)}}, \tag{3}$$

where $S_k^A(D)$ and $S_k^B(D)$ represent the observed performance scores of models $A$ and $B$, respectively, based on data $D$ up to stage $k$ (see Eq. 2), and $\hat{\sigma}_k^2$ is the pooled variance estimator.

At each interim analysis, the p-value $p_k$ is derived from $Z_k^{AB}$ and compared to a stage-adjusted significance threshold $\alpha_k$. Similarly, confidence intervals are constructed using an adjusted $\alpha$ to maintain appropriate reliability.

Deriving $p_k$ is the central challenge in sequential testing. If we were to apply the same critical value at each interim analysis as we would for a fixed-size test, the error rate would be inflated. The Pocock approach addresses this by using a constant critical value $c$ across all stages, such that the overall Type I error rate equals $\alpha$.

This method rests on several key assumptions: (i) the observations are independent, (ii) the test statistic $Z_k^{AB}$ follows an approximately normal distribution under the null hypothesis (as justified by the Central Limit Theorem when $T_k$ is sufficiently large), and (iii) $t$ is pre-specified. We note that other sequential methods may rely on different assumptions, for example relaxing the normality assumption nor and specifying pre-specified maximum number of analyses (Bibaut et al., 2024).

## 5.3 STOPPING RULES

Our framework incorporates multiple stopping criteria that we see as commonly practical or are commonly used in the sequential testing literature (Lewis, 2023; Rauch et al., 2020). We note that in principle, a user can stop for any reason, and the statistical guarantees will hold.

- **Efficacy Stopping:** Evaluation stops when the observed difference between models reaches statistical significance, and evaluating on more example will not aid the decision-making.

- **Equivalence Margin Stopping (Model Comparison):** Evaluation halts when the difference between models falls within a predefined equivalence margin, indicating that the models can be considered functionally equivalent. This is ideal for situations like model replacement, where the user wants to confirm that the new model is substantially better than the current.

- **Precision-Based Stopping (Single Model):** Evaluation terminates when the confidence interval around the model's estimated score is sufficiently narrow, aligning with the minimum detectable effect size (MDES). This ensures the model's performance is assessed with enough precision for practical decision-making, without unnecessary sampling.

- **Threshold Crossing Stopping:** Evaluation stops once the model's performance confidently exceeds or falls below a predefined threshold. This provides a definitive decision on whether the model meets or fails to meet a specified criterion, for example confirming it is faulty, avoiding further evaluation once a clear decision is made.

- **Futility Stopping:** Evaluation is stopped early if interim results suggest that achieving a practically meaningful difference is unlikely, thus preventing wasted resources.

- **Diminishing Returns Stopping:** Evaluation halts when the marginal gain in precision from additional samples falls below a specified threshold. This rule safeguards against excessive data usage once further sampling offers negligible improvements in reliability.

Each use case may call for a different stopping rule or a combination of rules. For instance, industry applications that require a minimal but robust performance difference may benefit from MDES-based precision stopping, whereas compute-constrained settings might prioritize efficacy stopping to reduce evaluation cost.

## 6 EXPERIMENTAL SETUP

Given adaptive calculation of significance (§5.1) and testing criteria (§5.3), we describe the technical implementation choice in our VLM benchmark experiments (§7). Specifically, we describe the data used (§6.1) and our implementation choices (§6.2).

## 6.1 DATA

We conduct our experiments using evaluation data from the *Open VLM Leaderboard* (Duan et al., 2024)[2], which, at the time of writing, includes 206 VLMs evaluated across 31 multimodal benchmarks. The leaderboard provides detailed metadata on the models, including their architectural components, underlying vision and language models, and model sizes. For a detailed list of the datasets see Appendix A.

For the benchmark's overall score, we adopt the standard unweighted mean-of-means as our metric. Since their released evaluation records provide only predictions, not scores, we utilize their LLM-as-a-Judge implementation with *Llama 3.1 8B*, to extract scores from the predictions.

## 6.2 ALGORITHM AND IMPLEMENTATION

In this section, we present the algorithm suggested for adaptive evaluation in model selection and its implementation details. The evaluation algorithm (for 2 models) proceeds as follows:

1. Let $b_{init}$ be the initial sample size, $b$ be the batch size, $A$ and $B$ the models we compare (see §5.2).
2. Generate predictions for batch $k$ using models $A$ and $B$.
3. Compute the evaluation scores over the data $N_k$: $S_A(N_k)$ and $S_B(N_k)$.
4. Acquire significance from the sequential testing algorithm.
5. Apply any mixture of the stopping rules defined in §5.1.
   (a) If a stopping condition is met, finalize the evaluation and return the computed score.
   (b) Otherwise, repeat the process from step 2.

Next, we describe implementation choices for our experiments. Utilizing the *gsDesign*[3] R package, which supports group sequential testing design and is widely used in clinical trials and medical research for efficient decision-making. We integrate this package into our Python pipeline, enabling seamless incorporation of group sequential design for model evaluation. The integration allows for adaptive evaluation with minimal computational and time overhead.

When comparing multiple models, we use pairwise comparisons. While pairwise comparisons introduce the challenge of multiple hypothesis testing and require additional correction methods, we simplify this aspect for the purpose of this study.

In all experiments, we configure our group algorithm with an initial sample size of 600 (100 per model), a batch size of 100 per iteration, a beta of 0.9, and the Pocock spending function as described in §5.1. These values were selected as reasonable defaults and were not tuned. From our experience in initial experiments, we expect them to work well enough for other settings, with an option for marginal gains upon hyperparameter search.

## 7 EXPERIMENTS AND RESULTS

To demonstrate the practical benefits of adaptive evaluation, we present a series of experiments that address common challenges in model evaluation. We quantify the efficiency gains for single-model score estimation (§7.1), show how adaptive stopping reduces the data required for reliable pairwise comparisons (§7.2), and illustrate the framework's applicability in real-world model development and deployment scenarios (§7.3).

## 7.1 SINGLE MODEL SCORE: APPROXIMATE ESTIMATION FOR COST-SAVING

In many real-world applications, users are often willing to accept a degree of uncertainty in exchange for faster evaluations and reduced costs. This experiment demonstrates how our framework enables

---

[2]Data sourced from the evaluation records available at https://huggingface.co/datasets/VLMEval/OpenVLMRecords, using commit *dbc5e10*.

[3]https://github.com/keaven/gsDesign

users to obtain an approximate ("ballpark") score for a model efficiently, by dynamically adjusting the number of samples used based on a user-defined confidence interval (CI) threshold.

We calculated the 95% CI half-width for each of the 206 models in the benchmark over 10 random seeds. Figure 1 shows the trade-off between the obtained CI and the number of examples. With our framework, users willing to accept a ±3-point CI will only need to run ≈ 15% of the samples, significantly reducing evaluation cost. If a ±2-point CI is required, around 30% of the samples are needed, and for a more precise ±1.5-point estimate, just about half of the samples will be necessary. Beyond this point, CI stabilizes due to diminishing returns: precision improves as $n^{-1/2}$ yielding rapid early gains but minimal impact from further sampling. A strong contributor to this effect is that in heterogeneous benchmarks, smaller datasets with higher variance dominate the unweighted mean of means. This caps overall uncertainty, and thus amplifies the effect of diminishing returns. For example, with 8,000 examples (55% of the dataset), the width of the confidence interval is 2.96 points. Expanding to the full 14,400 examples (100%) reduces the CI to 2.696 points—a mere 0.264-point difference. This marginal improvement (negligible in most practical cases) comes at the cost of nearly doubling the dataset size, underscoring the diminishing returns of additional sampling once a reasonable threshold is reached.

This highlights two key aspects: First, using the full benchmark is often wasteful, as reliability gains diminish beyond a certain point. Second, common heuristics Perlitz et al. (2024) – such as selecting 1,000 or fewer examples – tend to produce highly uncertain estimates, often exceeding ±5.5 points. Such large confidence intervals can render the results impractical for real-world decision-making. A secondary finding is that full benchmarks should be larger (echoing Perlitz et al. (2024)) and include large datasets to allow for reliable results. Specifically, the smallest and most varying datasets included in the benchmark are the ones most important to increase in size.

## 7.2 Pairwise Model Comparison

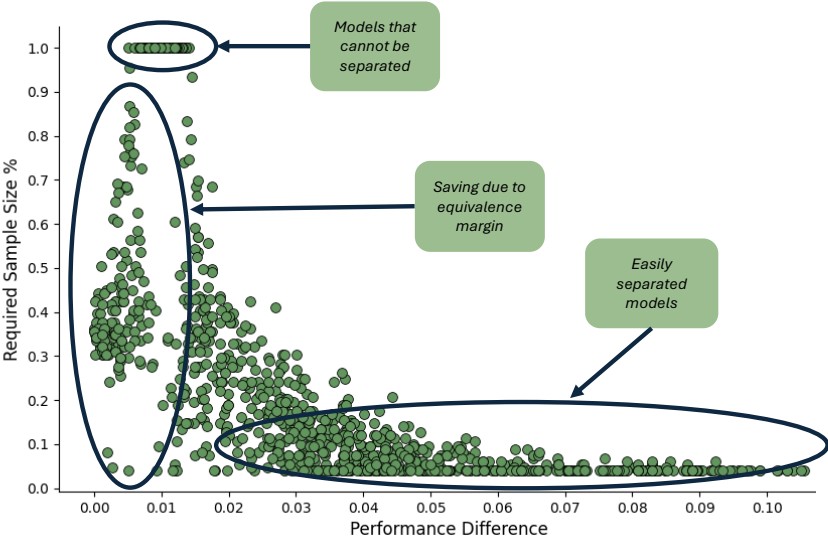

Figure 2: Required sample size (%) for model comparison with an equivalence margin of 2 points, as a function of performance difference. Each point represents a pairwise comparison between two models sampled from the top 50, evaluated sequentially. The y-axis shows the proportion of the full benchmark required to reach a confident decision, while the x-axis represents the performance difference between the models. As expected, larger performance differences require fewer samples, while smaller differences necessitate a larger portion of the dataset. Additionally, models with low performance differences that are inseparable without the equivalence margin can be distinguished when the margin is applied, leading to further sample savings.

When comparing two models, particularly those with subtle performance differences, practitioners often face the challenge of determining when enough evidence has been gathered to prefer one. This task becomes especially resource-intensive, and time-intensive when evaluating many models over large datasets. Our framework dynamically determines the minimum samples needed for statistically sound conclusions. Evaluation stops automatically when user-defined significance criteria are met, optimizing resource utilization.

To demonstrate the efficiency of our framework, we compare 1000 sampled pairs of models from the top 50 models in the benchmark. Evaluation was stopped as soon as the difference between the models reached statistical significance with 95% confidence.

We considered two distinct use cases in this experiment. In the first, the goal was strict comparison: we aimed to differentiate between the models and determine which one is definitively better. In the second use case, we introduced a relaxed comparison with an equivalence margin of 2 points. In this case, if the performance difference between the models was deemed to fall within 2 points, we stopped the evaluation and considered the models to be statistically equivalent. This approach is particularly useful for situations where small performance differences are of no practical importance or where the models are likely to be indistinguishable despite large sample sizes (in the latter futility stopping may also be useful).

The results, shown in Figure 7.2, reveal a clear and intuitive pattern: the number of examples required to confidently rank two models depends directly on the size of the performance gap. Larger differences can be detected with fewer examples, while smaller gaps require more data to achieve the same level of statistical confidence.

When performing strict comparisons, we observed that model pairs with a performance difference smaller than 1.2 points typically required the full dataset, as the evaluation failed to reach statistical significance with fewer samples ($\approx 24\%$). However, those might also be the least interesting comparisons as the comparison is futile and the models might be deemed practically equivalent. In contrast, for model pairs with differences larger than 2 points, our framework usually saved at least 60% of the evaluation effort, stopping early with high confidence.

In the more relaxed comparison with the 2-point equivalence margin, the number of examples required was indeed substantially reduced for pairs where the difference was close to zero. This is because the framework quickly identified that the models were within the predefined margin, and thus, further evaluation was unnecessary.

We compare our approach to a baseline where the user heuristically selects a fixed sample size of 200 examples per dataset, totaling 1,200 samples overall—a reasonable and practical guess for evaluation. For each model pair and seed, we assess whether this fixed sample size is sufficient to distinguish between the models with 95% confidence. To ensure a fair comparison, we use boot-straping for the fixed-size approach.[4] The results reveal a significant limitation of the fixed-sample approach: out of 1,000 model pairs, only 55% could be reliably differentiated with 95% confidence, as opposed to 76% by our framework. Most importantly, these failures are not visible to the user, unlike with our method.

This finding highlights the risk of arbitrarily selecting a sample size to reduce evaluation costs—while it may seem efficient, it is often just fast, but leading to the wrong conclusions. In contrast, our method dynamically determines the required number of examples, ensuring that evaluations are only as efficient as we can afford to be.

## 7.3 CASE STUDIES

Now that we've examined the experimental results in §7, we can return to the motivating use cases outlined earlier in §2. While these use cases are conceptually similar to the experiments we've conducted, we introduce additional rules for each scenario to highlight the flexibility of our adaptive evaluation framework. This demonstrates how users can mix and match different stopping rules (§5.3) based on their specific needs. In these experiments, we report the final sample savings and compare the baseline results to those achieved using a fixed sample size where applicable.

---

[4]Since the sample size is predefined, bootstrap is both applicable and statistically more powerful than our sequential test, offering better reliability per sample.

### 7.3.1 EFFICIENT EVALUATION WITH STATISTICAL GUARANTEES

A user with computational constraints seeks to evaluate models efficiently, but also report to higher-ups how confident they are in their estimations (reliability). In our experiment, this translates to accepting an equivalence margin of $\pm 2$ with $\alpha = 0.05$.

To simulate this, we rank five models sampled from the top 15 across 10 seeds, using pairwise comparisons. The performance gap between the best and worst models is at most five points, making differentiation challenging. Nonetheless, our method used only 60% of the examples, though 2.4 comparisons (out of 20) per run were indistinguishable and required the full benchmark.

### 7.3.2 MEANINGFUL CHANGE ACHIEVED

In production, teams must often ensure a new model outperforms the current one by at least 2 points before updating a model for deployment. We simulate this by sampling two models from the top 15, selecting one as a baseline, and comparing the other against it. This simulates the case of relatively small changes between model versions and performance. The stopping criteria chosen are a $\geq 2$-point improvement with 95% confidence or when such an improvement is deemed unlikely.

Unlike our pairwise ranking experiment, which tested differences in both directions, this approach is one-sided—we only check if the new model exceeds a predefined threshold for deployment. This can sometimes be a tougher test, because even if two models are different, it's not enough; the new model must show a specific, substantial improvement to justify deployment.

On average (over 100 random sampled pairs), our approach used only 63% of the examples compared to a fixed-sample approach.

### 7.3.3 MODEL DEVELOPMENT: EFFICIENT MODEL SELECTION.

To efficiently evaluate models while filtering out weak candidates, we apply two stopping rules. First, models scoring below 60 points are discarded immediately. Second, evaluation for higher-scoring models stops once their score is estimated within a $\pm 2$ point confidence interval.

This extends the single-model evaluation experiment by introducing an additional early-stopping rule, ensuring that only promising models undergo full evaluation while reducing unnecessary computation.

On average, running the benchmark across all models required only 30% of the total sample budget (over 10 seeds). 86 models were filtered out early as underperforming, while the rest stopped upon reaching the target CI. Without a threshold, the total sample usage was 50%, demonstrating the efficiency gained by filtering "faulty" models.

## 8 CONCLUSION

We introduce an adaptive evaluation framework that optimizes model benchmarking by balancing statistical reliability with computational efficiency. By leveraging sequential testing and adaptive stopping rules, our framework ensures that evaluations stop when practical and statistical needs are met, minimizing wasted resources without sacrificing the quality of results. We provide a systematic analysis of the trade-offs between efficiency and reliability in various evaluation scenarios, including model scoring, pairwise comparisons, and ranking tasks. This approach not only improves the resource efficiency of large-scale model evaluations but also enhances the transparency of the evaluation process, enabling users to make informed decisions based on clear trade-offs. Our framework paves the way for more efficient, robust, and flexible evaluation practices in both research and industry, addressing the growing challenges in benchmarking large language and vision-language models.

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

## A  DETAILED DATASET LIST

We use the following datasets from the *Open VLM Leaderboard* benchmark:

- **MMStarChen et al. (2024):** A multimodal benchmark designed to assess the capabilities of large VLMs across six core capabilities and 18 evaluation axes. It consists of 1,500 samples, each requiring visual comprehension and complex reasoning.

- **MMMUYue et al. (2023):** The *Massive Multi-discipline Multimodal Understanding benchmark*, containing questions spanning six disciplines: Art & Design, Business, Science, Health & Medicine, Humanities & Social Sciences, and Tech & Engineering. The benchmark requires college-level knowledge and reasoning. The leaderboard uses the validation split, which consist of 1050 examples, as the test set is private.

- **OCRBenchLiu et al. (2023b):** A benchmark for evaluating VLMs capabilities in optical character recognition (OCR). It covers five core tasks: Text Recognition, Scene Text-Centric Visual Question Answering (VQA), Document-Oriented VQA, Key Information Extraction, and Handwritten Mathematical Expression Recognition, with a total of 1K question-answer pairs.

- **AI2DKembhavi et al. (2016):** The AI2 Diagrams dataset, consisting of 3088 illustrative diagrams, in our test set, each accompanied by structured annotations and corresponding multiple-choice questions.

- **HallusionBench(Guan et al., 2024):** A benchmark designed to assess image-context reasoning in large VLMs, focusing on hallucinations and visual illusions. It includes 346 images paired with 1,129 questions. The test set include 951 sasmples in total.

- **MMBench(Liu et al., 2023a):** A benchmark of multiple-choice questions covering 20 different ability dimensions, such as object localization and social reasoning, for evaluating vision-language models. The questions are organized hierarchically into three levels: Perception and Reasoning (Level 1), Logic Reasoning (Level 2), and further subdivisions offering fine-grained assessments (Level 3). We use version $V1.1$. The test set include 7299 examples.

## B  FUTURE WORK

**Benchmark Building**  Benchmark builders often limit dataset sizes to manage evaluation costs, even when large-scale datasets are feasible through automated pipelines (Li et al., 2024). This limitation is exacerbated by the fact that full benchmarks should typically include larger datasets to provide reliable results, especially for close-performing models (echoing (Perlitz et al., 2024)). Our framework eliminates this tradeoff. By allowing the sample size to adjust based on user needs, benchmark creators can release larger, more comprehensive datasets without the concern of evaluation costs. This flexibility ensures that benchmarks can scale efficiently, supporting both standard model comparisons and more complex tasks requiring extensive data.

**Human Annotation**  Like in model evaluation, many methods for efficient human annotation rely on statistical and machine learning models, which require trust in their underlying assumptions and biases (Zouhar et al., 2025; Kossen et al., 2021). While these methods improve reliability per

sample, they often lack clear stopping criteria(Ashury-Tahan et al., 2024). Our framework can be directly applied to the annotation process, dynamically halting once sufficient statistical confidence is reached. Additionally, it can be integrated with existing methods, though this would require an examination and possible adjustment of the underlying assumptions to incorporate the stopping rules, which we leave to future work.

