# OpenReview forum: "Stop Guessing When to Stop Testing: Efficient Model Evaluation with Just Enough Data"
_ICLR.cc/2026/Conference — ICLR 2026 Conference Withdrawn Submission_

### Official Review · Reviewer_KQbJ · 2025-10-18

**Soundness:** 2
**Presentation:** 1
**Contribution:** 1
**Rating:** 0
**Confidence:** 4

**Summary:**

This paper adapts the sequential hypothesis testing procedure for evaluating large foundation models. Concretely, the authors make the claim that we should be using adaptive evaluation using the Pocock spending function (page 4). They describe what group sequential testing means for calculated frequentist p-values (pages 4-5) and explains the different possible stopping rules that exist (page 5). The authors evaluate their framework using a single chosen metric: an average of averages and obtain the scores with an LLM-as-a-judge methodology (page 6).  They show that sequentially testing a model on more data results in smaller performance difference (page 7),  perform an evaluation of two model comparisons and show how, relative to their methodology that they've established, stopping earlier can achieve cheaper results than running the full evaluation procedure. Overall, the paper positions itself as a framework for efficiently evaluating models by testing on the right amount of data by using the sequential hypothesis testing procedure and adaptive stopping rules.

**Strengths:**

A few different strengths I could point to:
1. The idea has not yet been exactly implemented, i.e. I have not been able to find a paper that does exactly this: stopping testing early on larger models by choosing how much data you need.
2. The idea itself is very clear as well.
3. The authors show a practically useful thing: using the full benchmark is often wasteful, since there is low reliability, and just selecting the number of points based on heuristics gives high uncertainty estimates (page 7).

**Weaknesses:**

Unfortunately, this paper has many more weaknesses than strengths. I will go through the most important ones here.

1. Writing is extremely poor. (i) There is no abstract (page 1), (ii)  key parts of the paper do not have any citations or references and feel just copy/pastes as outputs from a language model (just as one example, stopping rules at page 5); (iii) the flow of the paper is not well structured and the text is scattered throughout (especially pages 6-8). This makes the job of the reader very difficult.; (iv) an example of this writing is the mis-named tables/figurs (e.g. Figure 7.2 which does not exist in page 8, likely referring to the section and not the figure).

2. The idea seems like a very trivial and direct application of a very-well established idea.

2a) That is, it seems that the authors just took the group sequential testing and stopping rule idea and applied it to language models. They claim that this is a "framework to optimize sample efficiency", but this framework is not your contribution -- the application of this framework on LLMs is. That is, if an analyst uses this framework on tabular data, they have also not "designed a new framework" - they have simply applied it. Therefore, this seems to be a major misunderstanding of the nature of the contribution -- you have not developed the framework; you've provided an application.

2b) You claim that you provide a "systematic analysis of the trade-offs between efficieancy and reliability in varous evaluation scenarios" (page 8). Where is that "systematic analysis"? I understand you evaluate it on all the models (206 models, page 7); but it seems: (i) you have used only one dataset; (ii)  you evaluate simply on the basis of using your procedure vs evaluating everything all-together, which seems to be a very narrow comparative scope?

2c) When defending the claim "why this is useful" (section 2), it's not sufficient to just cite one paper like Perlitz et al 2024 and assume now it's obvious why this framework is useful. It is not obvious. We've been doing evaluations on LLMs and large models very well up until now. Is the paper making the claim that all these evaluations are unnecessary, i.e. all the well-established benchmarks can be significantly made cheaper? If so, this is not obvious and we need a much higher burden of proof for this.

3. I also think the framing of the paper is misleading. Consider the title: "STOP GUESSING WHEN TO STOP TESTING: EFFICIENT MODEL EVALUATION WITH JUST ENOUGH DATA". No one *really* guesses when to stop testing -- there are no scenarios when a bunch of people are sitting around a model and providing guesses on whether we should stop the test now. I understand this is intended as a rhethorical device, but, at best, it is very poorly positioned, and at worst, very misleading.

**Questions:**

1. You claim you aim to eschew fixed-size evaluation. Are you making the claim that the current evaluation benchmarks are poor? Specifically:

1a) Should we abandon benchmarks like MMLU, ARC, ARC2, used to evaluate models? If this is a part of the claim of the paper, could you explain and provide a stronger defense why the current benchmarks used widely should be abandoned and give some empirical evidence to support this? If we should not abandon these benchmarks, could you clarify which benchmarks are you referring to, exactly?

1b) You claim that we should eschew fixed-sive evaluations (page 1). Do you think fixed-size evaluations have no role at all? If so, can you give a larger critique of this field -- this is a very big claim. If not, can you adjust the claim?

2. Do you agree you do not propose a framework (page 1) but, rather, "adapt" a framework to evaluating LLMs? How do you think this affects your novelty?

3. What's the goal of listing out all the stopping rules in page 5 with such great detail?

4.  You claim that "such as selecting 1,000 or fewer examples – tend to produce highly uncertain estimates, often exceeding ±5.5 points".Is there an analysis of this claim in the paper?  If so, where can I find this?

5. A big part of your evaluation framework is the difference between "typical" evaluation and your evaluation. In a typical evaluation, you generate predictions and evaluate. In your framework, you also need to perform additional steps: acquire significance from the sequential testing algorithm and apply the stopping rules (page 6). Waht is the overhead of this in terms of: (a) clock-time; (b) computational/token count; (c) overall speed?

6. Your current writeups in the case studies are just written out words: it's very difficult to follow. Could you summarize the main claims in a figure/table and highlight what's necessary?

---

### Official Review · Reviewer_Ai8z · 2025-10-22

**Soundness:** 3
**Presentation:** 3
**Contribution:** 2
**Rating:** 6
**Confidence:** 4

**Summary:**

The method proposes an empirical analysis of existing group sequential testing methodologies for LLM evaluation, and shows interesting insights on the achievable efficiency gains when using these methods to stop evaluation early when evaluation is expensive.

**Strengths:**

1. The problem is extremely important. Evaluations are indeed expensive
2. Efficiency gains are actually possible, which is an interesting finding worth reporting.
3. The work is extremely simple and focused (even though I'm sure plenty of work went into getting there). And I don't mean it in a bad sense - simplicity is oftentimes **underrated**.

**Weaknesses:**

1. In the top-50, we will probably get quite a lot of pairs of a strong and weak model, which means we'll stop early quite often. But in practice, when a model is gradually improved, the effect size of the improvement is not that big. The work doesn't appear to demonstrate usefulness the more practical scenario (maybe by taking top-10, which are pretty close). The fact that most models in the benchmark are easy to separate is not very insightful in practice.
2. There are plenty of related works on always-valid inference which aren't cited as related work.
3. There is no abstract. Perhaps the authors forgot it?

**Questions:**

NA

---

### Official Review · Reviewer_boz3 · 2025-10-27

**Soundness:** 1
**Presentation:** 2
**Contribution:** 2
**Rating:** 2
**Confidence:** 3

**Summary:**

In this work, the authors propose a statistically grounded solution: an adaptive evaluation framework based on sequential testing.
The description of the proposed algorithms are supposed to be presented in Section 5. However, the Section 5 only seems to give a high level schematic outline of their algorithm. Because of lack of details, it is hard to see the value of this work.

**Strengths:**

The idea of sequential testing seems interesting.

**Weaknesses:**

Because the paper does not provide sufficient details on its algorithm and justification, it is hard to evaluate its impact.

**Questions:**

The overall writing should be improved significantly. In particular, as a statistician, I found the content in Section 4 very vague, below the standard of technical writing. There are many necessary components unexplained, e.g., the data generation mechanism (i.e., the underlying model) is not explained at all. This is not acceptable.

---

### Official Review · Reviewer_6PPk · 2025-10-29

**Soundness:** 2
**Presentation:** 3
**Contribution:** 2
**Rating:** 2
**Confidence:** 4

**Summary:**

This paper proposes an adaptive evaluation framework for VLM ) based on group sequential testing. Instead of using fixed-size benchmarks, the framework stops evaluation when statistical and practical criteria are met.

**Strengths:**

Originality:
- Novel use of group sequential testing for VLM evals
- Interesting taxonomy of stopping rules to VLM evals

Quality:
- Good use of statistical foundations linked to evals
- Experiments demonstrate consistent savings across multiple scenarios

Clarity:
- Well-written paper that’s easy to follow. Good link between the theory and practical side of the paper
- Good balance between theory and practice

Significance:
- Addresses an important problem of eval costs and more efficient evals

**Weaknesses:**

- Limited experiments: only Open VLM leaderboard & no evidence despite claims that this works for LLMs or other modalities. Significant overclaim here unless LLMs and other modalities shown to justify the generality

- Insufficient baselines - the baselines are mentioned but no comparison to Polo et al. (2024), Zhang et al. (2024), Zhao et al. (2024)

- No abstract to the paper

**Questions:**

- Unclear empirical gain: when should we use sequential testing vs dataset sampling

- Can you compare with more baselines?

- Hyperparameter justification: Why specific values chosen? Please provide sensitivity analysis or principled selection criteria.

- Were there cases where sequential testing performed poorly or required more samples than fixed-size? When does the method fail?
e.g. What happens with very small datasets in benchmark?  What happens if the score distribution is skewed? What if stopping criteria conflict?

---

### Note · Authors · 2025-11-13

I have read and agree with the venue's withdrawal policy on behalf of myself and my co-authors.